# A Customizable and Low-Cost Ultraviolet Exposure System for Photolithography

**DOI:** 10.3390/mi13122129

**Published:** 2022-12-01

**Authors:** David Eun Reynolds, Olivia Lewallen, George Galanis, Jina Ko

**Affiliations:** 1Department of Bioengineering, University of Pennsylvania, Philadelphia, PA 19104, USA; 2Department of Biomedical Engineering, Boston University, Boston, MA 02215, USA; 3Department of Pathology and Laboratory Medicine, University of Pennsylvania, Philadelphia, PA 19104, USA

**Keywords:** UV exposure system, photolithography, droplet microfluidics, low-cost, portable

## Abstract

For microfluidic device fabrication in the research, industry, and commercial areas, the curing and transfer of patterns on photoresist relies on ultraviolet (UV) light. Often, this step is performed by commercial mask aligner or UV lamp exposure systems; however, these machines are often expensive, large, and inaccessible. To find an alternative solution, we present an inexpensive, customizable, and lightweight UV exposure system that is user-friendly and readily available for a homemade cleanroom. We fabricated a portable UV exposure system that costs under $200. The wafer holder’s adjustable height enabled for the selection of the appropriate curing distance, demonstrating our system’s ability to be easily tailored for different applications. The high light uniformity across a 4” diameter wafer holder (light intensity error ~2.9%) was achieved by adding a light diffusing film to the apparatus. These values are comparable to the light uniformity across a 5” diameter wafer holder from a commercial mask aligner (ABM 3000HR Mask Aligner), that has a light intensity error of ~4.0%. We demonstrated the ability to perform photolithography with high quality by fabricating microfluidic devices and generating uniform microdroplets. We achieved comparable quality to the wafer patterns, microfluidic devices, and droplets made from the ABM 3000HR Mask Aligner.

## 1. Introduction

Photolithography is a process of optical techniques for patterning a photosensitive polymer. Common polymers used for photolithography include epoxy-based photoresists that have high optical transparency [1,2]. The two most common versions include negative and positive photoresist. For negative photoresist, selectively exposing the photo-reactive areas to ultraviolet (UV) light through a chrome or mylar mask transfers the mask pattern onto the polymer, while the unexposed areas disintegrate. The opposite is true for positive photoresist [3]. Because this microfabrication technique allows for customized surface fabrication, it is frequently used for the development of sensor technologies, general electronics, and microfluidics which utilize patterned substrates [4,5,6,7,8,9,10,11,12]. However, this operation requires an expensive mask aligner or UV lamp systems for UV exposure, making it difficult to build a homemade cleanroom that can perform photolithography in common laboratory settings. 

UV mask aligners and commercial UV lamp exposure systems, while precise and conveniently automated, can be large, complex, and costly, ranging from $16,000 (Eveflow)–$40,000 (BlackHole Lab). Thus, this paper aims to create a portable, low-cost, and customizable UV light system that can easily be applied in different laboratory settings. Thus far, several groups have sought to address this challenge as well. For example, Naggay et al. have developed several self-built UV lamp exposure systems [13,14]. Their first model focused on building a benchtop system that was focused on LED power and design [13]. The second system, though, concentrated on integrating a scaled-down laminar hood for working in a dust-free environment [14]. Other groups have worked on developing and optimizing their photolithography systems for achieving high aspect channel ratios, reaching sub-nanometer features, multi-layer devices, large circular, illumination areas, and increasing exposure rounds [15,16,17,18,19]. However, because the majority of these systems remain expensive and non-portable, they remain incompatible for homemade cleanrooms. To produce a more convenient system, specifications regarding budget, size, and available materials must be considered.

Herein, we present the Diffuser UV (DUV) Light System that is fabricated with desktop 3D printers and over-the-counter materials. The DUV Light System is composed of an assembly of two major components, the UV lamp containment unit and wafer holder. The containment unit contains shelves that are designed for the optimal curing of different photoresist deposited wafer conditions. Adjustable supports have been explored before [15]; however, our design takes advantage of the adjustability inside the container itself rather than requiring separate parts. The light diffusing film adequately disperses the UV light across the wafer holder, ensuring the light uniformity is comparable to commercial equipment. The weight of the DUV Light System (2.75 kg) also makes the system portable and compact. Together, the combination of these different features supports the DUV Light System’s ability to be a customizable, accessible, low-cost, and portable alternative to commercial mask aligners or UV lamp exposure systems for photolithography.

## 2. Materials and Methods

### 2.1. UV Lamp Containment Unit

To properly cure photoresists for different applications, the distance between the UV lamp and the silicon wafer was made adjustable. This inspired the design of a container with 1 cm thick shelves where the UV lamp and wafer holders can be raised or lowered. The shelves themselves were also designed to be 1 cm apart to identify the ideal exposure height.

The DUV Light System components are outlined in Table 1. More specifically, our DUV Light System was fabricated with a desktop 3D printer (Creality Ender 5, Shenzhen, China). For 3D printing material, the holder was printed with black Polylactic Acid (PLA) (HATCHBOX, Pomona, CA, USA) at 100% infill to prevent the passage of outside light. The dimensions of the unit were 20.32 cm (width) × 25.4 cm (length) × 20.32 cm (height) (Appendix A). The UV lamp containment unit ended up weighing 2.25 kg. To block out light from the area of wafer and photomask removal and insertion, a black-out curtain was fitted to the 3D-printed unit with magnetic tape (Master Magnetics, Castle Rock, CO, USA) and black fabric (Sedona Design, Inc., Sedona, AZ, USA).

### 2.2. UV Light Holder

For UV light exposure, the UV lamp needs to have a wavelength between 300–450 nm. More specifically, because negative SU-8 is used in this study, the wavelength requires a 365 nm wavelength according to the curing specifications of SU-8 2000 and SU-8 3000 [20,21]. Therefore, the lamp that was chosen was a 50W 365 nm UV lamp (Everbeam, Surrey, CA, USA).

Because the lamp is 16.00 cm (width) × 20.57 cm (length), the UV light holder was designed to be 19.68 cm (width) × 24.38 cm (length) (Appendix A). Additionally, the thickness of the UV light holder was designed to be 1 cm in thickness, identical to the shelves of the containment unit, to make height adjustments of the UV lamp straightforward. The UV light holder was fabricated with a desktop 3D printer (Stratasys F170, Eden Prairie, MN, USA). Similarly, the UV lamp containment unit was printed with PLA at 100% infill. The resulting weight was 0.14 kg. To optimize the timing accuracy of curing from the UV lamp, the lamp was plugged into an outlet timer (BN-LINK, Santa Fe Springs, CA, USA). Additionally, a light diffusing film (Edmund Optics, Barrington, IL, USA) was taped underneath the UV lamp for exposure uniformity.

### 2.3. Wafer Holders

The top and bottom wafer holders are used for positioning and securing the wafer to the photomask for UV exposure, respectively. Similarly to the UV lamp, the thickness of the top and bottom holders were designed to be 1.0 cm in thickness. The dimensions of the bottom and top wafer holders were 19.58 cm (width) × 24.38 cm (length) (Appendix A) and 16.51 cm (width) × 16.51 cm (length) (Appendix A), respectively. Additionally, 3” and 4” wafers marking were designed into the bottom wafer holder for wafer and UV lamp centering. 5” × 5” markings were also integrated into the bottom wafer holder for chrome mask securement and alignment between the UV light and wafer. For both wafer holders, through 0.64 cm holes were designed for alignment and securing a wafer and mylar mask between the two parts.

To maintain fabrication consistency, the wafer holders were also 3D printed with the Stratasys F170 using black PLA at 100% infill as material. Upon assembly, 0.635 cm × 3.81 cm screws (Hillman, Cincinnati, OH, USA) and corresponding wing nuts (Hillman, Cincinnati, OH, USA) were used to properly secure the wafer holders together. Together the two holders weighed approximately 0.5 kg.

### 2.4. UV Exposure Calibration

To calculate the necessary distance for curing between the silicon wafer and the UV lamp, a UV intensity meter (General Tools, New York, NY, USA) was used. By using this sensor, the intensity of the UV light over a period of time at different heights was collected. Locating and taking the integral of the trendline between the data points verified the required time and exposure energy for SU-8 curing. Not only is the desired time for the curing process based on the total UV intensity and distance, but is also based on the thickness of the SU-8; the height of the wafer holder can be adjusted to account for this factor according to exposure data for SU-8.

To verify the intensity uniformity across the wafer with and without the diffusion film, a UV intensity meter was used. Measurements were collected at 17 points on the wafer after 20 s had elapsed. The points selected included a point at the center of the bottom wafer holder, 8 evenly distributed points around the center at a radius of 3.175 cm, and 8 evenly distributed points around the center at a radius of 4.45 cm. These data were collected and analyzed in the form of a UV intensity map.

### 2.5. Microfluidic Droplet Device Fabrication and Testing

The mylar photomasks were designed in 2D AutoCAD and fabricated by Fineline Imaging. The DUV Light and ABM 3000HR Mask Aligner devices (h = 50 µm) were made using photolithography with SU-8 3050. For the DUV Light wafers, the chosen UV curing specifications included a 5 cm distance between the wafer and UV light. Polydimethylsiloxane (PDMS) devices were then given inlet and outlet ports for dispensing and collecting water-in-oil droplets. Subsequently, the devices were bonded to glass slides with plasma bonding for 30 s at low power (Harrick Plasma Cleaner, PDC-32G, Ithaca, NY, USA). The channels were finally treated with 1% trichloro (1H, 1H, 2H, 2H perfluorooctyl) silane in Droplet Generation Oil (BioRAD, Hercules, FL, USA) for making the channels hydrophobic.

After the fabrication and treatment process with both UV light systems, the T-junction droplet generating device from both wafers were used for droplet validation. The performance of the devices was tested using varied flow ratios of water to oil (BioRad, Hercules, FL, USA) to form uniform droplets. The droplet formation procedure was tested with oil-to-water flow ratios between 5:1 (1:0.2 mL/h) to 10:1 (2:0.2 mL/h). The best oil-to-water flow ratios and their respective results were reported in the Results section.

## 3. Results & Discussion

### 3.1. DUV Light System Fabrication and Assembly

3D printing is convenient for its rapid prototyping and inexpensive material. In our case, the UV containment box, wafer and UV light holders were fabricated through 3D printing. Because we wanted to expedite the 3D printing process, two separate 3D printers were used. Subsequently, the printed parts were sanded to remove edges and layer lines. However, because the resolutions were different between the 3D printers, extra sanding was required. From printing to assembly (Figure 1A), the total required time was 72 h.

Additional setup requirements were required for the system’s functionality. For instance, to increase high light uniformity, a light diffuser sheet was taped underneath the UV light (Figure 1B). Screws and nuts were used for securing the top and bottom wafer holders for photomask and wafer contact (Figure 1C). A notch was designed into the bottom mask for the passage of the UV intensity meter wiring, which was used for optimizing for exposure times (Figure 1D). Additionally, to ensure the interior of the containment box was fully enclosed from outside light, a black fabric was attached to the UV containment box (Figure 1E). For rapidly attaching and removing the black curtain to the walls of the UV containment box, magnetic tape was added to its exterior edges. Together, the combination of these different components supported the system’s functionality and cost-effectiveness.

### 3.2. DUV Light System Validation

The device itself is designed to be adjustable for various applications. To outline the specifications of adjustment for different purposes, calculating curing times and verifying the quality of the UV light exposure with the UV intensity meter was crucial (Appendix A). Measuring the UV intensity for 120 s (Figure 2A) resulted in an intensity drift, which is indicative of lower exposure energies require less exposure time. For calculating curing times (Figure 2B), because we were aiming for 50 µm height in cured SU-8 negative photoresist, a 250 mJ/cm^2^ exposure energy was chosen [21]. The combination of results enabled for the creation of a user-friendly chart to simplify the process of finding the proper curing time according to the user’s desired application (Appendix A). Because our minimum feature widths are between 30–40 µm and the height is 50 µm, the maximum aspect ratio was not assessed. For SU8-3050, the maximum thickness that can be reached is a little over 100 µm, thus, it can be assumed the maximum aspect ratio that can be achieved is 3:1.

To verify the curing times, UV intensity uniformity data was collected both with and without light diffusing film. This data is represented as intensity maps in Figure 2C,D. Between the two figures, it is shown that the beam uniformity inside a 4” diameter was improved by the light diffusing film addition to the apparatus, narrowing the light intensity error from ~5.7% to ~2.9%. This is comparable to the beam uniformity across the 12.7 cm wafer of the ABM 3000HR Mask Aligner, which is ~4.0% [22]. Thus, the film became a necessity in the device apparatus. Other materials and systems were considered as well, including a fresnel lens array or a collimator. Comparatively, though, fresnel lens arrays ($2000) and collimators ($200) can be quite expensive. Not to mention, because the DUV UV light’s area is over 500 cm^2^, many collimators would be required to cover the entire surface area. This would increase the overall costs of the DUV system. However, because fresnel lens array and collimator have exact light focusing and customized integrated properties, they are recommended for sub micron features with complex edges

### 3.3. Comparison of Developed Patterns

A photomask for droplet microfluidics was designed to compare the resolution and quality of our UV light system to the ABM 3000HR Mask Aligner. The reported resolutions from the DUV Light System ended up being 30 ± 5 µm based on critical dimensions around 30 µm not being consistently uniformly patterned. Using the optimized set up described previously, the new photomask was used to create the droplet microfluidic devices. The PDMS devices created from the DUV Light System (Figure 3A) were comparable to the ABM 3000HR Mask Aligner microfluidic device (Figure 3B) as shown by the clarity of the smaller features in the droplet and T-junction microfluidic device. Not to mention, the edge quality of the produced microstructures remained similar, as highlighted by cross-sections of the T-junction microfluidic devices (Appendix A). For the T-junction device, the water (vertical) and oil (horizontal) channels were designed to be 50 µm and 100 µm, respectively. Between the two reported T-junction devices, there were no differences in dimensions, supporting the DUV Light System’s ability to quickly prototype accurate droplet microfluidics PDMS devices.

### 3.4. Microfluidic Droplet Device Performance Testing

The droplet microfluidic devices produced through the DUV Light System and ABM 3000HR Mask Aligner were then used to make droplets (Figure 4A,C). The devices were then given inlet and outlet ports for dispensing and collecting water-in-oil droplets. Subsequently, the devices were bonded to glass slides with plasma bonding. The oil-to-water flow speed ratio was tested at 5:1, 6:1, 7:1, 8:1, 9:1, and 10:1 to determine the finest speed for uniform droplet formation. The optimal flow ratio was determined to be 5:1 at an oil and water speed of 1 mL/h and 0.2 mL/h, respectively (Appendix A). This ratio produced uniformly sized droplets (d = 100 µm) that created the desired monolayer (Figure 4B,D). To confirm uniformly sized droplets, the coefficient of variation was calculated between the DUV Light System and ABM 3000HR Mask Aligner. The coefficient of variation was 2.6% and 2.7% for the DUV Light System and ABM 3000HR Mask Aligner, respectively (Appendix A).

## 4. Conclusions

The DUV Light System presented in this paper addresses the need for a user-friendly, affordable, mobile, and efficient UV light system for photolithography UV light exposure. More specifically, we present a UV light system whose fabricated costs are under $200, weighs 2.75 kg, and requires only 72 h for fabrication and assembly. With the limited materials and instruments used for achieving these features, we were not only able to fabricate wafers with indistinguishable patterns to an ABM 3000HR Mask Aligner, but were able to produce identical uniform droplets.

While there are many clear advantages to our DUV Light System, there also remain some setbacks. For example, compared to the ABM 3000HR Mask Aligner, resolutions range between 500 nm to 1 µm, however, the DUV Light System’s resolutions are constrained to 30 ± 5 µm. Not to mention, the DUV Light System assembly was conducted through two separate desktop 3D printers. Nevertheless, the presented UV light system efficiently serves for the rapid fabrication and prototyping of droplet microfluidic systems. It can also be used to create microchannels, microwells, and other patterns that do not require a sub-micron resolution. Thus, providing a customizable, accessible, inexpensive, and lightweight alternative to commercial ultraviolet mask aligners or UV lamp exposure systems for photolithography.

## Figures and Tables

**Figure 1 micromachines-13-02129-f001:**
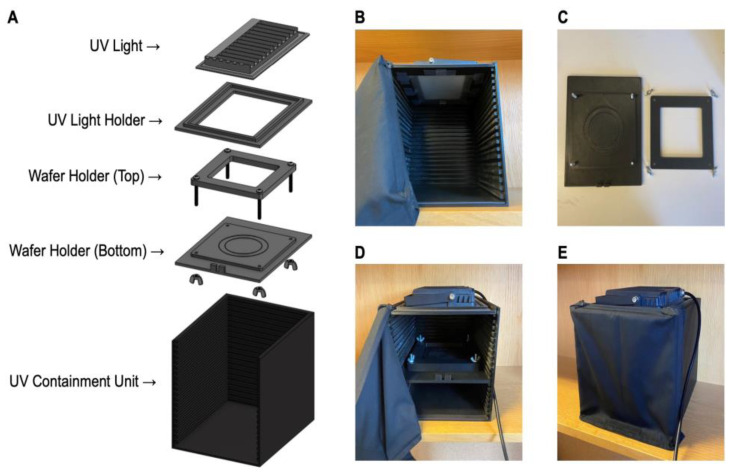
Schematic of the DUV Light System. (**A**) 3D exploded view of the DUV Light System. (**B**) Photo of the inside of the UV light system, showcasing the shelving system and the light diffusing film underneath the UV lamp. (**C**) Photo of the individual mask holder components. (**D**) Photo of the completed assembly. (**E**) Photo of the assembly with the curtain closed, as one would set up for use of the device.

**Figure 2 micromachines-13-02129-f002:**
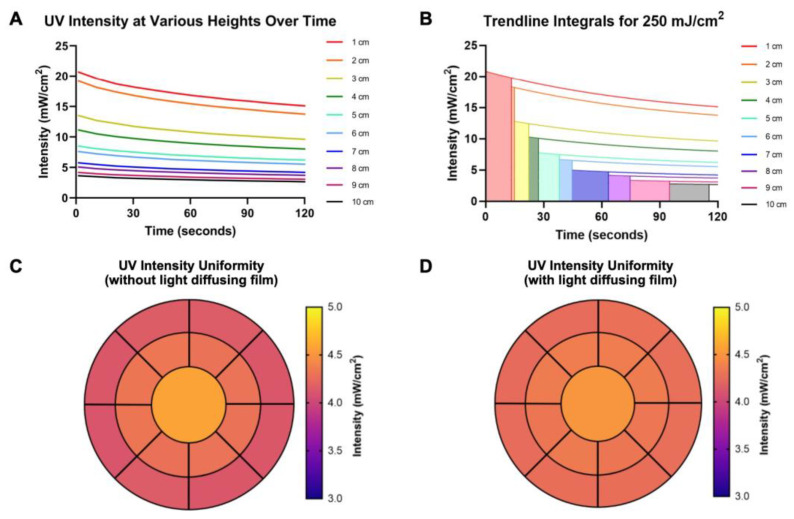
DUV Light System Performance Validation. (**A**) Graph of the light intensity from the UV lamp over a 2 min period. (**B**) Graph of the light intensity trendlines featuring the area under each curve corresponding to 250 mJ/ cm^2^ exposure time. (**C**) UV light intensity map without the light diffusing film. (**D**) UV beam intensity map with the light diffusing film.

**Figure 3 micromachines-13-02129-f003:**
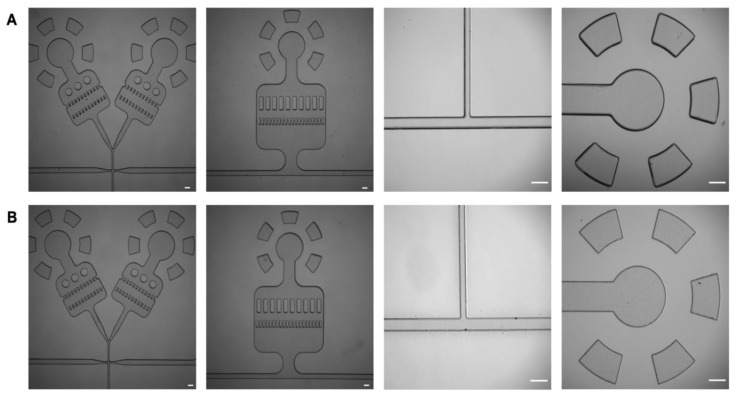
PDMS droplet microfluidic devices. (**A**) Images of DUV Light System wafer patterns, including PDMS flow focusing and T-junction microfluidic devices. (**B**) Images of ABM 3000HR mask aligner wafer patterns, including PDMS flow focusing and T-junction microfluidic devices. (Scale bar = 150 µm).

**Figure 4 micromachines-13-02129-f004:**
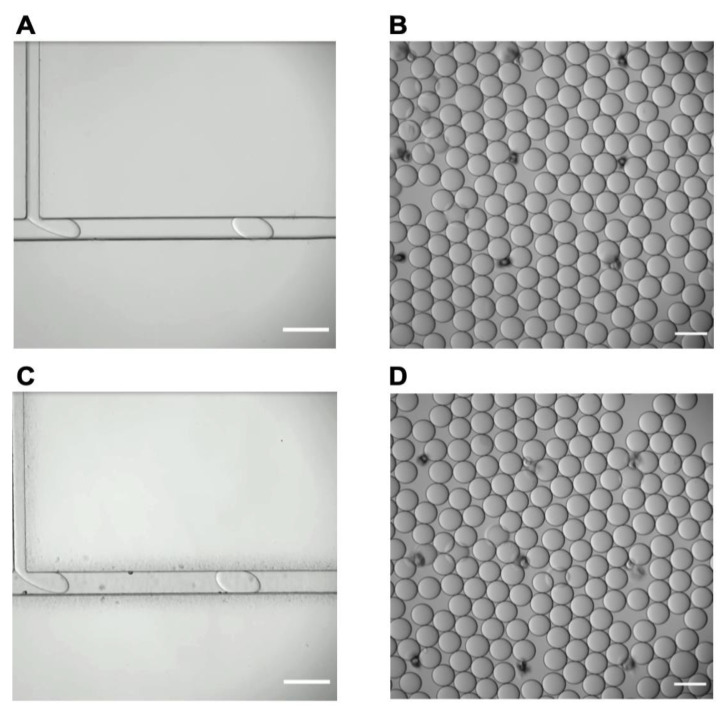
Droplet formation. (**A**) DUV Light System derived T-Junction microfluidic device at a 5:1 oil to water flow rate ratio with (**B**) derived droplets. (**C**) ABM 3000HR Mask Aligner derived T-Junction microfluidic device at a 5:1 oil to water flow rate ratio (**D**) with derived droplets. (Scale bar = 150 μm).

**Table 1 micromachines-13-02129-t001:** DUV Light System Components.

Item	Quantity	Vendor	Cost ($)
3D Printer PLA	2	HATCHBOX	49.98
Magnetic Tape	1	Master Magnetics	7.69
Black Fabric	1	Sedona Design, Inc.	7.95
UV Lamp	1	Everbeam	48.99
Outlet Timer	1	BN-LINK	30.99
Light Diffusing Film	1	Edmund Optics	45.00
Screws & Wing Nuts	4	Hillman	5.36
Total			$195.96

## Data Availability

Not applicable.

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
