# Peer review of "A Customizable and Low-Cost Ultraviolet Exposure System for Photolithography"

_micromachines, 2022, doi:10.3390/mi13122129_

Round 1

Reviewer 1 Report

In this manuscript, Ko et al developed a cost-effective, customizable, and portable UV exposure system, and then applied it to perform photolithography for the fabrication of microfluidic device. Overall, I think this work is interesting, valuable, and offers a practical alternative UV exposure system for photolithography, which will attract attention across the wide readership in Micromachines as well as the researchers working in the fields of microfluidics. Therefore, I recommend its publication with minor revision as noted: 1) In Figure 3, the authors described the dimensions of the device; I suggest an additional description of the depth of device since it has influence on the droplet diameters; 2) In Figure 4, histograms of the droplet diameters are suggested to added for better comparison of the droplet quality; and 3) Some spelling mistakes needed to be checked: for example, in the caption of Figure 3, the (B) is missed before “Images of ABM…”, and in Section 2.2, “For UV light exposure, the UV lamp needs to have a wavelength between 300-450 nm.3” Does the superscript number “3” represent a reference?

Reviewer 2 Report

The authors present a low-cost exposure system for SU-8 or other photoresists and show proof-of-concept with soft-lithography of a droplet-microfluidic system. The presented system is not novel and several similar studies have been presented before, e.g.:

Micromachines | Free Full-Text | Hydrogel Patterns in Microfluidic Devices by Do-It-Yourself UV-Photolithography Suitable for Very Large-Scale Integration | HTML (mdpi.com)

https://aip.scitation.org/doi/10.1063/1.5035282

It would be good if the authors can comment on these papers and discuss what sets their system apart from these other custom-made setups.

Considering the extremely low price of the setup I still think that the presented study holds some interest for the reader. I would suggest including the following information:

1. Add stp or stl files in the supplementary to directly print the parts.

2.  Quantify the homogeneity of the UV light. The given graphical representations only allow a rough estimation. What is the homogeneity in % over the exposure area?

3. Can the homogeneity be improved e.g., by adding a Fresnel lens array or collimator instead of a diffusor film?

4. The authors give no information of the edge quality of the produced microstructures. It would be good to include a cross-section/cut of the channels. I would assume that the light is not very collimated leading to rounded edges.

5. It would be good to include the suppliers in table 1.

6. What is the maximum SU-8 thickness/aspect ratio that can be processed with the exposure system?

Round 2

Reviewer 2 Report

All my comments are adressed. I suggest publication.